# Certified Adversarial Robustness via Mixture-of-Gaussians Randomized Smoothing

**Vaughn Rostermundt**
Mathematics Department
California Polytechnic State University
San Luis Obispo, CA 93407
vrosterm@calpoly.edu

**Brendon G. Anderson**
Mechanical Engineering Department
California Polytechnic State University
San Luis Obispo, CA 93407
bga@calpoly.edu

## Abstract

We propose a generalization of randomized smoothing (RS) that uses noise drawn from a mixture of $K$ Gaussians. We prove that, under a mild Lebesgue integrability condition on the base classifier, the proposed method is decomposable into any one of $K!$ equivalent, $K$-step sequential applications of standard RS. We leverage this multitude of decompositions to show that the mixture-of-Gaussians smoothed classifier inherits Lipschitz continuity from the strongest Lipschitz bound amongst its standard RS constituents. Consequently, we prove that the $\ell_2$-certified radius of the proposed method is inherited from the largest certified radius of its constituents; the mixture-of-Gaussians smoothed model is at least as robust as smoothing with each of the Gaussians individually. CIFAR-10 experiments show that the proposed model exhibits comparable clean accuracy (i.e., zero attack radius) and maximum certified radius to those of standard RS using its maximum-variance constituent, while significantly improving certified accuracy at intermediate attack radii.

## 1 Introduction

Due to their impressive capabilities in a variety of challenging data-driven tasks, machine learning models are seeing increased deployment in safety-critical applications, such as self-driving vehicles [1, 2] and power grid operations [3]. As such, it is imperative to ensure that these models operate reliably, even in the presence of unreliable data such as adversarial attacks, i.e., imperceptible manipulations in the input data that are maliciously designed to cause system failures [4, 5, 6].

One of the most popular methods for defending against adversarial attacks is *randomized smoothing* (RS), which drowns out the effects of data manipulation by classifying based on the average—or most probable—prediction of a collection of intentionally corrupted, noisy variants of the input [7, 8, 9]. In general, altering models to enhance robustness, for instance via RS, comes at the expense of some predictive accuracy, inducing an *accuracy-robustness tradeoff* [10, 11]. Various methods have been introduced to push the Pareto frontier of accuracy and robustness, including by combining randomized smoothing with adversarial training [12, 13, 14], optimizing the noise distribution used by RS [15], and interpolating at test time between different models that are pretrained to either optimize accuracy or robustness [16]. However, achieving high levels of robustness via RS relies on using high-variance noise, which can result in over-aggressive smoothing and lead to decision region shrinkage—a detriment to classwise accuracy [17]. On the other hand, RS with small-variance noise can maintain high accuracy, but the resulting robustness tends to be lackluster.

In this work, we generalize RS to incorporate a *range* of variances in the smoothing noise. Specifically, we propose to smooth models using mixtures of Gaussians with differing variances. To our surprise, this idea has found minimal exploration in the literature, despite being a rather obvious "next step" to

39th Conference on Neural Information Processing Systems (NeurIPS 2025) Workshop: Reliable ML from Unreliable Data.

build off standard RS that uses a single Gaussian. To the best of our knowledge, the only other work that explicitly considers RS with mixtures-of-Gaussians is tucked away in Appendix B.1 in Eiras et al. [18]. However, their certified radius is proven based on Lipschitzness of the smoothed classifier $\overline{g}$, rather than on the much stronger Lipschitzness of $\Phi^{-1} \circ \overline{g}$ (the composition with the inverse standard normal cumulative distribution function). This results in their certified radii of robustness being significantly limited, as they are upper-bounded by $\frac{1}{\sqrt{2\pi}}$. We prove a much stronger Lipschitz guarantee, and consequently much larger certified radii (Theorem 2) that enjoy unbounded growth as the model becomes more confident. Also related is Lyu et al. [19], which consecutively applies Gaussian smoothing to the input in an adaptive fashion (not a mixture of Gaussians), but their certified radii scale with $\frac{1}{\sqrt{\sum_{k=1}^{K} \frac{1}{\sigma_k^2}}}$, whereas ours takes a stronger form, scaling with $\max_{k \in \{1,...,K\}} \sigma_k$.

## 1.1 Summary of Contributions

In this paper, we achieve the following contributions (with all proofs deferred to Appendix A):

1. We formalize a new variant of RS that uses a mixture-of-Gaussians smoothing distribution.
2. We prove that, under a mild Lebesgue integrability condition, the proposed method is equivalent to sequential application of standard RS with a single Gaussian.
3. We leverage the sequential representation of our method to prove that the model inherits the strongest Lipschitz bound amongst its standard RS constituents, and we use this bound to prove an $\ell_2$-certified radius.
4. CIFAR-10 experiments show that the model's clean accuracy and maximum certified radius rival standard RS using the maximum-variance Gaussian alone, and that the certified accuracy of our model significantly increases at attack radii between these two extremes.

## 2 Proposed Method

Let $f \colon \mathbb{R}^d \to \{1, \ldots, n\}$ be a pretrained $n$-class base classifier defined by

$$f(x) = \arg\max_{i \in \{1,...,n\}} g_i(x),$$

where $g \colon \mathbb{R}^d \to \mathbb{R}^n$ is the underlying soft classifier. Often, the score vector $g(x)$ is a probability vector in the simplex $\Delta^{n-1} := \{y \in [0,1]^n : \sum_{i=1}^n y_i = 1\}$ (e.g., when the model applies softmax to the logits), which we will ultimately assume to hold in our certificates of Theorem 1 and Theorem 2 that follow. We propose to smooth the base classifier using a mixture of Gaussians. To do so, let $\mu_1, \ldots, \mu_K \in \mathbb{R}^d$ be $K$ mean vectors, and let $\Sigma_1, \ldots, \Sigma_K \in \mathbb{R}^{d \times d}$ be $K$ positive semidefinite covariance matrices. Our mixture-of-Gaussians distribution is taken to be the $K$-fold convolution

$$\mathrm{MoG} := \mathcal{N}_1 * \cdots * \mathcal{N}_K,$$

where every $\mathcal{N}_k := \mathcal{N}(\mu_k, \Sigma_k)$ is a normal distribution with mean $\mu_k$ and covariance $\Sigma_k$. Thus, $\epsilon \sim \mathrm{MoG}$ if and only if $\epsilon = \epsilon_1 + \cdots + \epsilon_K$ with independent $\epsilon_k \sim \mathcal{N}_k$ for all $k$. Our proposed smoothed classifier is defined by

$$\overline{f}(x) = \arg\max_{i \in \{1,...,n\}} \overline{g}_i(x), \quad \overline{g}(x) = \mathbb{E}_{\epsilon \sim \mathrm{MoG}}[g(x + \epsilon)].$$

This form of smoothing, expressed in terms of expectations of the model's probability vector, has been popularized in recent years and is known as *soft smoothing* [20, 13, 21]. In cases where the base classifier has codomain $\Delta^{n-1}$, soft smoothing recovers *hard smoothing*, as originally formulated in [8], as the entropy of the model's probability vector decreases to zero (which can be enforced in practice by using a temperature-scaled softmax) [12]. In the case that $K = 1$, $\mu_1 = 0$, and $\Sigma_1 = \sigma^2 I_d$ for some $\sigma > 0$, our method reduces to standard RS using a single isotropic Gaussian.

## 3 Analysis and Certification of the Model

A key property underlying our mixture-of-Gaussians smoothing method is that it can be viewed as a sequential application of standard RS, in any order, under a Lebesgue integrability assumption.

**Assumption 1.** It holds that $\mathbb{E}_{\epsilon \sim \text{MoG}}[\|g(x + \epsilon)\|] < \infty$.

**Proposition 1.** *Let $\sigma \colon \{1, \ldots, K\} \to \{1, \ldots, K\}$ be an arbitrary permutation. If Assumption 1 holds, then the following sequential relationship holds:*

$$h_1(x) := \mathbb{E}_{\epsilon_{\sigma(1)}}[g(x + \epsilon_{\sigma(1)})],$$
$$h_2(x) := \mathbb{E}_{\epsilon_{\sigma(2)}}[h_1(x + \epsilon_{\sigma(2)})],$$
$$\vdots$$
$$\overline{g}(x) = h_K(x) := \mathbb{E}_{\epsilon_{\sigma(K)}}[h_{K-1}(x + \epsilon_{\sigma(K)})].$$

Assumption 1 is mild in practice. For instance, when $g$ has codomain $\Delta^{n-1}$, as is the case for models with softmax at the output, we see that $\mathbb{E}_{\epsilon \sim \text{MoG}}[\|g(x + \epsilon)\|] \leq 1 < \infty$, so Assumption 1 holds.

As a consequence of our model being equivalent to sequential standard smoothing, we obtain strong Lipschitz continuity guarantees (Theorem 1), as well as certified radii (Theorem 2):

**Theorem 1.** *Assume that $g$ has codomain $\Delta^{n-1}$, suppose that $\Sigma_k = \sigma_k^2 I_d$ for all $k \in \{1, \ldots, K\}$, and let $\Phi \colon \mathbb{R} \to [0, 1]$ denote the cumulative distribution function of $\mathcal{N}(0, 1)$. It holds for all $i \in \{1, \ldots, n\}$ that $x \mapsto \Phi^{-1}(\overline{g}_i(x))$ is $L$-Lipschitz continuous with $L = \min_{k \in \{1, \ldots, K\}} \frac{1}{\sigma_k}$.*

**Theorem 2.** *Assume that $g$ has codomain $\Delta^{n-1}$ and suppose that $\Sigma_k = \sigma_k^2 I_d$ for all $k \in \{1, \ldots, K\}$. Let $x \in \mathbb{R}^d$. It holds that $\overline{f}(x + \delta) = y := \overline{f}(x)$ for all $\delta \in \mathbb{R}^d$ such that*

$$\|\delta\|_2 \leq r(x, y) := \frac{\max_{k \in \{1, \ldots, K\}} \sigma_k}{2} \left( \Phi^{-1}(\overline{g}_y(x)) - \Phi^{-1}\left(\max_{y' \neq y} \overline{g}_{y'}(x)\right) \right). \tag{1}$$

*Remark* 1. Theorem 2 recovers the certified radius in Cohen et al. [8, Theorem 1] when $K = 1$.

Theorem 1 shows that mixture-of-Gaussians smoothing results in a model that is at least as smooth (as measured by the Lipschitz constant) as the model would be if it were smoothed using any one of the Gaussians on their own. On the other hand, Theorem 2 takes a bit more care to fully dissect. Roughly, the result can be interpreted as mixture-of-Gaussians smoothing giving rise to certified radii at least as large as those generated by standard smoothing with the maximum-variance Gaussian alone. However, the certified radii of mixture-of-Gaussians smoothing can, in theory, be *strictly* stronger than those obtained by simply performing standard RS with the maximum-variance Gaussian. The intuition is this: the Gaussian random vector $\epsilon_k \sim \mathcal{N}(0, \sigma_k^2 I_d)$ concentrates near the boundary of the zero-centered $\ell_2$-ball of radius $\sigma_k \sqrt{d}$ [22, Theorem 3.1.1]. Therefore, by including noise from $\mathcal{N}(0, \sigma_k^2 I_d)$ with $\sigma_k < \max\{\sigma_1, \ldots, \sigma_K\}$ in the smoothing procedure, we are including in the average score calculation additional predictions generated by inputs within a closer vicinity of $x$, specifically, within the *interior* of the ball $\{x' \in \mathbb{R}^d : \|x' - x\|_2 \leq \max\{\sigma_1, \ldots, \sigma_K\}\sqrt{d}\}$. These noisy samples that are closer to $x$ are likely to have classification probabilities that are similar to the baseline $g(x)$ (due to the continuity of most practical base classifiers $g$). Therefore, if the base model's prediction, as encoded by $g(x)$, is correct in the first place (which is typically the case for many well trained base classifiers), then the close samples help boost the confidence in that prediction, resulting in an increased margin $\overline{g}_y(x) - \max_{y' \neq y} \overline{g}_{y'}(x)$, and thereby increase the certified radius as evidenced by the form of (1).

These theoretical insights of the certified radius of Theorem 2 suggest that there are possible benefits to "filling out" the interior of the ball over which the (majority of the probability mass of the) smoothing operation is being carried out, rather than simply concentrating the smoothing noise on the ball's boundary as is done in standard RS. Interestingly, smoothing with entirely "filled out" uniform distributions over $\ell_p$-norm balls has been explored in the prior work Yang et al. [23]. However, their certified $\ell_2$-radius (their Theorem S.1) takes a more complicated form than ours, it is bounded above, and it is numerically shown to yield close, but slightly smaller certified radii than standard Gaussian RS on CIFAR-10. Conversely, our certified radius formula is a natural and intuitive extension of the well-known radius from standard RS [8, Theorem 1], it enjoys unbounded growth as the model becomes more confident (i.e., as $\overline{g}_y(x)$ approaches 1), and we find that our formula together with our mixture-of-Gaussians approach for filling out the ball's interior manifests in significant certified accuracy increases in our CIFAR-10 experiments that follow.

## 4    Experiments and Discussion

As is standard in the RS literature, we evaluate our certified accuracies against a range of $\ell_2$-attack radii (i.e., at a given attack radius, we compute the percentage of the test set with certified radius at least as large as the attack). We conduct our experiments on the CIFAR-10 dataset by modifying the open-source codebase Cohen et al. [24] (that implements standard RS Cohen et al. [8] with a fixed, pre-trained base classifier) to sample smoothing noise from a mixture of Gaussians rather than a single Gaussian, and we utilize our certified radius formula (1). We implement our method with mixtures of zero-mean, isotropic Gaussians, i.e., $\mathcal{N}_k = \mathcal{N}(0, \sigma_k^2 I_d)$, so that our robustness certificates in Theorem 1 and Theorem 2 hold. We evaluate certified accuracy curves for varying numbers of Gaussians, $K$, as well as varying ranges of standard deviations, $\sigma_k$. The $K$ standard deviations are uniformly spaced apart. We remark that standard RS corresponds to $K = 1$.

Figure 1 shows the results for varying Gaussians counts, $K$. It is observed that increasing $K$ increases the certified accuracy, with significant improvements found at larger attack radii. This finding corroborates the theoretical discussion following Theorem 2; using more Gaussians permits the samples to "fill out" the $\ell_2$-sphere that is being averaged over which tends to increase confidence, thereby boosting certified radii. It can also be observed that the "endpoints" of the mixture-of-Gaussians certified accuracy curves are quite close to those of the standard RS curve corresponding to the maximum-variance Gaussian ($\sigma = 0.5$ in the left plot, $\sigma = 1$ in the right plot). That is, mixture-of-Gaussians smoothing has comparable clean accuracy (left endpoint) and maximum certified radius (right endpoint) to standard RS with the maximum-variance Gaussian, yet the certified accuracies at intermediate radii are seen to significantly increase. This suggests that, instead of performing standard RS at some chosen variance, one may significantly benefit robustness by introducing lower-variance Gaussians to the smoothing scheme, at the possible expense of small decreases in accuracy.

Figure 2 shows the results for varying standard deviation intervals. We see that the certified accuracy curves are primarily controlled by the maximum variance, with relatively little effect due to the minimum variance. This suggests that the mixture-of-Gaussians boost in the certified radii could be mainly attributed to "filling in" an annulus near the surface of the maximum-variance $\ell_2$-sphere, and less attributed to including samples very close to the input. Explicitly teasing out the effects of sampling from various regions of the input space poses an interesting direction for future research.

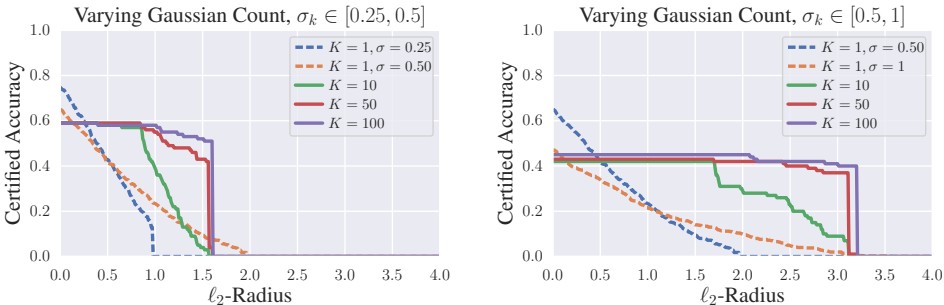

Figure 1: Certified accuracies for various Gaussian counts $K$, compared with standard RS ($K = 1$). Left plot uses standard deviations in the interval $[0.25, 0.5]$, right plot uses the interval $[0.5, 1]$.

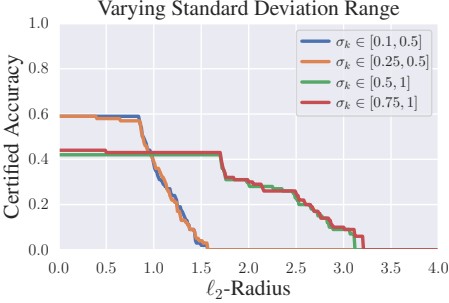

Figure 2: Certified accuracies for various standard deviation intervals, with $K = 10$.

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

## A    Proofs

**Proposition 1.** *Let $\sigma\colon \{1,\ldots,K\} \to \{1,\ldots,K\}$ be an arbitrary permutation. If Assumption 1 holds, then the following sequential relationship holds:*

$$h_1(x) := \mathbb{E}_{\epsilon_{\sigma(1)}}[g(x + \epsilon_{\sigma(1)})],$$
$$h_2(x) := \mathbb{E}_{\epsilon_{\sigma(2)}}[h_1(x + \epsilon_{\sigma(2)})],$$
$$\vdots$$
$$\overline{g}(x) = h_K(x) := \mathbb{E}_{\epsilon_{\sigma(K)}}[h_{K-1}(x + \epsilon_{\sigma(K)})].$$

*Proof of Proposition 1.* Since all of the relationships are definitions except for $\overline{g}(x) = h_m(x)$, this is the only equality that needs to be proven. Suppose that Assumption 1 holds. Then, Fubini's theorem applies to the expectations under consideration, so we find that

$$
\begin{aligned}
h_K(x) &= \mathbb{E}_{\epsilon_{\sigma(K)}}[h_{K-1}(x + \epsilon_{\sigma(K)})] \\
&= \mathbb{E}_{\epsilon_{\sigma(K)}}[\mathbb{E}_{\epsilon_{\sigma(K-1)}}[h_{K-2}(x + \epsilon_{\sigma(K)} + \epsilon_{\sigma(K-1)})]] \\
&\vdots \\
&= \mathbb{E}_{\epsilon_{\sigma(K)}}\mathbb{E}_{\epsilon_{\sigma(K-1)}}\cdots\mathbb{E}_{\epsilon_{\sigma(2)}}[h_1(x + \epsilon_{\sigma(K)} + \epsilon_{\sigma(K-1)} + \cdots + \epsilon_{\sigma(2)})] \\
&= \mathbb{E}_{\epsilon_{\sigma(K)}}\cdots\mathbb{E}_{\epsilon_{\sigma(1)}}[g(x + \epsilon_{\sigma(K)} + \cdots + \epsilon_{\sigma(1)})] \\
&= \mathbb{E}_{\epsilon_{\sigma(K)},\ldots,\epsilon_{\sigma(1)}}[g(x + \epsilon_{\sigma(K)} + \cdots + \epsilon_{\sigma(1)})] \\
&= \mathbb{E}_{\epsilon \sim \mathrm{MoG}}[g(x + \epsilon)] \\
&= \overline{g}(x).
\end{aligned}
$$

$\square$

**Theorem 1.** *Assume that $g$ has codomain $\Delta^{n-1}$, suppose that $\Sigma_k = \sigma_k^2 I_d$ for all $k \in \{1,\ldots,K\}$, and let $\Phi\colon \mathbb{R} \to [0,1]$ denote the cumulative distribution function of $\mathcal{N}(0,1)$. It holds for all $i \in \{1,\ldots,n\}$ that $x \mapsto \Phi^{-1}(\overline{g}_i(x))$ is $L$-Lipschitz continuous with $L = \min_{k \in \{1,\ldots,K\}} \frac{1}{\sigma_k}$.*

*Proof of Theorem 1.* We may assume that every $\mu_k = 0$ without loss of generality, since, for $\epsilon_k \sim \mathcal{N}_k$, we may always absorb the mean into the input $x$ as follows: $\mathbb{E}_{\epsilon_k}[g(x + \epsilon_k)] = \mathbb{E}_{\epsilon_k' \sim \mathcal{N}(0,\sigma_k^2 I_d)}[g(x + \mu_k + \epsilon_k')] = \mathbb{E}_{\epsilon_k' \sim \mathcal{N}(0,\sigma_k^2 I_d)}[g(x' + \epsilon_k')]$, where $x' = x + \mu_k$.

Let $k \in \{1, \ldots, K\}$ be arbitrary, and let $\sigma \colon \{1, \ldots, K\} \to \{1, \ldots, K\}$ be a permutation satisfying $\sigma(K) = k$ (which obviously exists). By Proposition 1, the following decomposition holds:

$$h_1(x) := \mathbb{E}_{\epsilon_{\sigma(1)}}[g(x + \epsilon_{\sigma(1)})],$$
$$h_2(x) := \mathbb{E}_{\epsilon_{\sigma(2)}}[h_1(x + \epsilon_{\sigma(2)})],$$
$$\vdots$$
$$\overline{g}(x) = h_K(x) := \mathbb{E}_{\epsilon_k}[h_{K-1}(x + \epsilon_k)].$$

Therefore, $\overline{g}$ is the model obtained through standard RS applied to the function $h_{K-1}$, using the smoothing distribution $\mathcal{N}(0, \sigma_k^2 I_d)$. Hence, Zhai et al. [12, Lemma 1] gives that $x \mapsto \Phi^{-1}(\overline{g}_i(x))$ is $\frac{1}{\sigma_k}$-Lipschitz continuous for all $i \in \{1, \ldots, n\}$. Since $k$ is arbitrary, the result follows. $\qquad\square$

**Theorem 2.** *Assume that $g$ has codomain $\Delta^{n-1}$ and suppose that $\Sigma_k = \sigma_k^2 I_d$ for all $k \in \{1, \ldots, K\}$. Let $x \in \mathbb{R}^d$. It holds that $\overline{f}(x + \delta) = y := \overline{f}(x)$ for all $\delta \in \mathbb{R}^d$ such that*

$$\|\delta\|_2 \le r(x, y) := \frac{\max_{k \in \{1, \ldots, K\}} \sigma_k}{2} \left( \Phi^{-1}(\overline{g}_y(x)) - \Phi^{-1}\left(\max_{y' \ne y} \overline{g}_{y'}(x)\right) \right). \qquad (1)$$

*Proof of Theorem 2.* This follows from the same argument used in the proof of Zhai et al. [12, Theorem 2], with the Lipschitz constant $L = \min_{k \in \{1, \ldots, K\}} \frac{1}{\sigma_k} = \frac{1}{\max_{k \in \{1, \ldots, K\}} \sigma_k}$ from Theorem 1. $\qquad\square$

