# OpenReview forum: "Certified Adversarial Robustness via Mixture-of-Gaussians Randomized Smoothing"
_NeurIPS.cc/2025/Workshop/Reliable_ML — NeurIPS 2025 - Reliable ML Workshop_

### Official Review · Reviewer_LMNn · 2025-09-06
**Interesting idea with room for more discussion**

**Rating:** 8
**Confidence:** 3

**Review:**

# Summary
Randomized smoothing (RS) is a technique used to make classifiers robust to adversarial noise in the input by ensuring similar prediction across a neighbourhood of the given input. However, fixing a low/high variance for the Gaussian smoothing noise results in low robustness/accuracy respectively. This paper considers, instead, sampling the smoothing noise from a convolution of Gaussians with different variances.  They show an equivalence between this method and sequential application of standard RS.  Using this equivalence, they prove this results in a classifier whose Lipschitz constant and certified radius depends on the largest variance. The method improves on both accuracy and robustness (as measured by certified radius) over standard RS on the CIFAR dataset.

# Strengths
1. The intuition regarding concentrating smoothing noise on the boundary vs the interior of the ball around a given input is novel, helpful, and well-explained.
2. The writing is clear and precise, with assumptions made explicit.
3. The equivalence of the authors' method with sequential application of standard RS seems novel, though I am not an expert on the topic.

# Weaknesses
1. It seems nonstandard to use the term ‘mixture of Gaussians’ to describe a convolution (sum) of Gaussian random variables. In my understanding, a Gaussian mixture model typically samples first from a categorical over *K* Gaussians, then from the chosen Gaussian. Eiras et al [18] also use the term ‘mixture of Gaussians’ in a similar way, though there’s a coefficient in front of each Gaussian (I believe Eiras et al [18] and the authors’ formulations should be equivalent though). The chosen method seems better described by a sum (or convolution) of Gaussians.
2. The experiments include only the original randomized smoothing baseline, and not any of the other related approaches mentioned in the paper.

# Suggestions
1. A convolution of Gaussians is itself a Gaussian whose mean and variance is the sum of constituent means and variances. What do the standard bounds from previous literature yield for this resulting Gaussian, if applied naively? It could be useful to include an explicit comparison of how the authors’ decomposition-based analysis improves on this.
2. Would the authors’ results generalise to any Gaussian that can be decomposed as a convolution of lower-variance Gaussians?
3. In the experiments, could the authors please include more recent randomized smoothing baselines, as discussed in the paper?

# Ethics
N/A

---

### Official Review · Reviewer_e8Ci · 2025-09-20
**RS method with interesting methodology but missing writing sections**

**Rating:** 5
**Confidence:** 3

**Review:**

This paper provides a methodology to defend against adversarial attacks via randomized smoothing. This work generalizes prior approaches to incorporate a range of variance in smoothing noise with an aim in reducing the accuracy-robustness tradeoff.

Pros:
- Well thought out and novel solution with strong mathematical backing (both in main text and appendix)
- Prior work discussion is well done with clear draws as to why the authors' method is superior.
- The experiment uses a known dataset and compares a standard RS model to their solution with clear plots and explanations.

Cons:
- I wish the comparison plots included some of the other methods discussed in the introduction. I believe you do better than the simplest approach, but I'd be curious as to how it stacks up against methods that combine RS with adversarial training, optimized noise distribution, etc.
- The paper is missing some key sections. The experiment section is tied in with the discussion making it difficult to separate interpretation versus experimental results. There are no limitations and no conclusion sections. Generally, I think this paper needs more structure. It ends suddenly with no real takeaway for the reader.
- While the takeaways from the experiment are clearly that your model is better than standard, the experiment itself was difficult to parse through.